# Additional energy scale in $SmB_6$ at low-temperature

L. Jiao[1], S. Rößler[1], D.J. Kim[2], L.H. Tjeng[1], Z. Fisk[2], F. Steglich[1,3,4] & S. Wirth[1]

Topological insulators give rise to exquisite electronic properties because of their spin-momentum locked Dirac-cone-like band structure. Recently, it has been suggested that the required opposite parities between valence and conduction band along with strong spin-orbit coupling can be realized in correlated materials. Particularly, $SmB_6$ has been proposed as candidate material for a topological Kondo insulator. Here we observe, by utilizing scanning tunnelling microscopy and spectroscopy down to 0.35 K, several states within the hybridization gap of about $\pm 20$ meV on well characterized (001) surfaces of $SmB_6$. The spectroscopic response to impurities and magnetic fields allows to distinguish between dominating bulk and surface contributions to these states. The surface contributions develop particularly strongly below about 7 K, which can be understood in terms of a suppressed Kondo effect at the surface. Our high-resolution data provide insight into the electronic structure of $SmB_6$, which reconciles many current discrepancies on this compound.

[1] Max-Planck-Institute for Chemical Physics of Solids, Nöthnitzer Str. 40, 01187 Dresden, Germany. [2] Department of Physics and Astronomy, University of California, Irvine, CA 92697, USA. [3] Center for Correlated Matter, Zhejiang University, Hangzhou 310058, China. [4] Institute of Physics, Chinese Academy of Sciences, Beijing 100190, China. Correspondence and requests for materials should be addressed to S.W. (email: wirth@cpfs.mpg.de).

In the past few years, the concept of strong topological insulators, which exhibit an odd number of surface Dirac modes characterized by a $\mathbb{Z}_2$ topological index, has attracted great interest. In this context, it was theoretically predicted that some Kondo insulators, such as $SmB_6$, $Ce_3Bi_4Pt_3$, CeNiSn, $CeRu_4Sn_6$, are candidates for strong three-dimensional (3D) topological insulators[1,2]. In particular, $SmB_6$ is intensively studied because of its simple crystal structure and clear signatures of a Kondo hybridization gap. Theoretically, a common picture of the multiplet $f$-states and the Kondo hybridization effect is shared among different band structure calculations for $SmB_6$ (refs 2–7), as sketched in Fig. 1 (Supplementary Fig. 1). Because of strong spin-orbit coupling and crystal field effects, the $f$-states of Sm are split into several multiplets as presented in Fig. 1a. Considering the symmetry of the multiplets, only the $\Gamma_7$ and $\Gamma_8^{(1)}$ bands are allowed to hybridize with the Sm $d$-band via the Kondo effect[4,6]. As a result, two hybridization gaps (denoted as $\Delta_1$, $\Delta_2$) may open at different energies, as sketched in Fig. 1b (in principle only $\Delta_2$ is a well-defined gap). Although topological surface states (TSS) are unambiguously predicted to reside within the hybridization gap[2–7], no consensus has been reached on the structure of the TSS around the Fermi energy ($E_F$). Nonetheless, the prediction of TSS provides an attractive explanation for the four decades-old conundrum[8] of $SmB_6$, which exhibits a plateau in the resistivity typically below about 5 K (refs 9,10).

Experimentally, the existence of metallic surface states below about 5 K has been best illustrated by electrical transport measurements on $SmB_6$ (refs 10–12). However, the origin of these surface states and their topological properties remain

controversial, in spite of intensive investigations. Several properties of $SmB_6$ interfere with a straightforward interpretation. One major issue arises with respect to the size of the hybridization gap. Spectroscopic measurements observed a large hybridization gap of about 15–20 meV (refs 13–24), which is normally understood by considering a single $f$-band hybridizing with a conduction band via the Kondo effect (Supplementary Fig. 1). Typically, additional features within this energy scale are assumed to be in-gap states. In some cases, the in-gap states are further ascribed to TSS (refs 15,17). On the other hand, analyses of thermal activation energies derive a small excitation energy of 2–5 meV, which shows bulk properties and is understood in terms of a small, likely indirect, bulk gap[25–27] or in-gap states[28–30]. Obviously, different probes, as well as different ranges in the measurement temperatures reveal only either the bigger or the smaller hybridization gap sketched in Fig. 1b. Nevertheless, these measurements provide essential constraints to the sizes of the two hybridization gaps. In terms of topology (that is, trivial or non-trivial surface states), experimental results, even obtained by using the very same method, are conflicting among many reports[14–24,31–34]. Considering the exotic phenomena, which appear only within $\pm 20$ meV and below 5 K, measurements with very high-energy resolution and at very low-temperature are highly desired.

Another severe difficulty, which contributes to such a wide discrepancy among the experimental results, is caused by the surface itself. Specifically, the (001) surface of $SmB_6$ is polar[23]. This can induce different types of band bendings[14], quantum well confinements[35], charge puddles and surface reconstructions[36–39]. Specifically the latter may give rise to conducting surface layers on its own[23]. Frequently, different types of surfaces (B- and Sm-terminated, reconstructed and non-reconstructed) coexist at different length scales on one and the same cleaved surface, which may complicate interpretation of spectroscopic results, for example, by angle-resolved photoemission spectroscopy (ARPES).

We therefore conduct scanning tunnelling microscopy/spectroscopy (STM/STS) down to the base temperature of 0.35 K with an energy resolution of about 0.5 meV. This allows us to identify the fine structure of the hybridization gaps on large and non-reconstructed surfaces in the sub-meV scale. Moreover, by measuring the impurity, magnetic-field and temperature dependence of the STS spectra, we are able to attribute bulk and/or surface contributions to these states, and unveil a new energy scale of $\simeq 7$ K, which provides an important piece of the puzzle for a unified picture of $SmB_6$.

## Results

**Topography and STS spectra at base temperature.** $SmB_6$ crystallizes in a cubic structure with a lattice constant $a = 4.133$ Å, Fig. 2a. The topography of a non-reconstructed surface, presented in Fig. 2b, exhibits clear atomic resolution. Here, the distance of about 4.1 Å and the arrangement of the corrugations is in good agreement with the cubic structure of $SmB_6$. The very small number of defects compared with the number of unit cells within the field of view ($>5,200$) not only indicates high sample quality but also ensures that the measured spectrum is not influenced by defects. The absence of any corrugation other than along the main crystallographic axes, as nicely seen in the inset of Fig. 2b, clearly indicates a B-terminated surface[37,39].

The differential tunnelling conductance $g(V) \equiv dI(V)/dV$, measured at $T = 0.35$ K and far away from any impurity, exhibits several anomalies close to $E_F$, marked by (i)–(v) in Fig. 2c. A change in the slope of $g(V)$ around $\pm 20$ meV, suggests a pronounced loss of local density of states within this energy

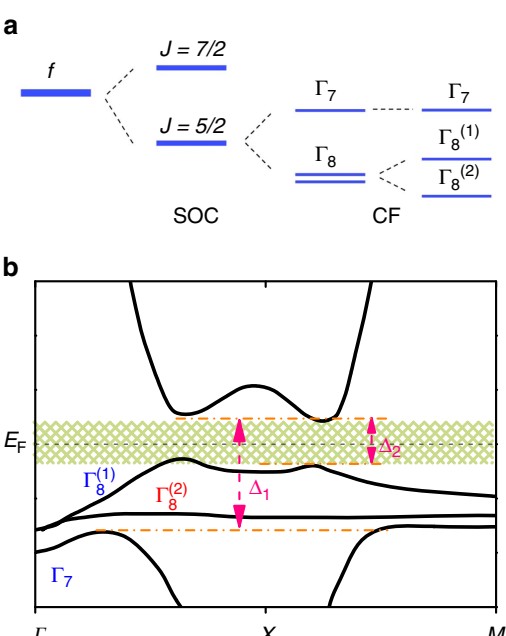

**Figure 1 | Sketch of the multiplet $f$-states and the resulting band structure. (a)** Evolution of energy levels of the $f$-states in $SmB_6$, which follows from the work of refs 6,7. The $f$-states are split into $J = 7/2$ and $J = 5/2$ states by spin-orbit coupling (SOC). The $J = 5/2$ state, which is slightly below $E_F$, is split into a $\Gamma_7$ doublet and a $\Gamma_8$ quartet by the crystal field (CF). Away from the $\Gamma$ point, the $\Gamma_8$ quartet is further split into $\Gamma_8^{(1)}$ and $\Gamma_8^{(2)}$ doublets. **(b)** A schematic bulk band structure of $SmB_6$ based on calculations of refs 2–7. Kondo hybridization between the $\Gamma_7$, $\Gamma_8^{(1)}$ bands and the conduction band opens two gaps which are denoted as $\Delta_1$ (typically around 20 meV) and $\Delta_2$. The shaded area marks the small bulk gap which may host in-gap states.

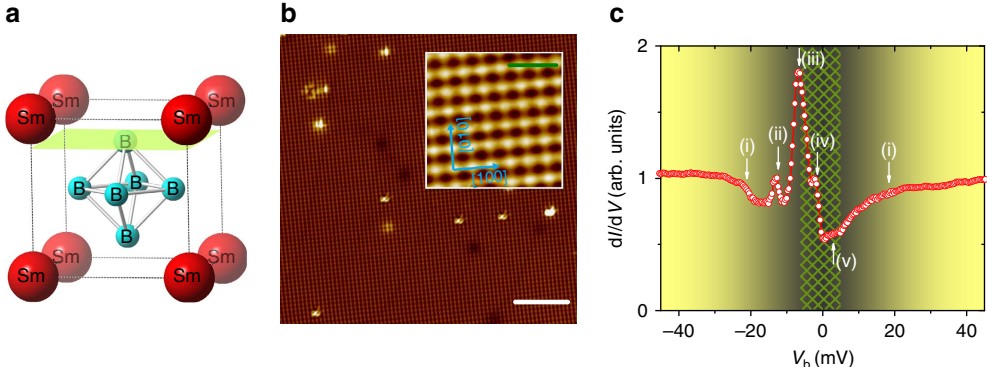

**Figure 2 | B-terminated surface and STS spectra at base temperature.** (**a**) Cubic crystal structure of $SmB_6$ with lattice constant $a = 4.133$ Å. The green plane indicates a cleave with B-terminated surface. (**b**) STM topography on a $30 \times 30\,nm^2$ non-reconstructed B-terminated surface of $SmB_6$ ($T = 0.35$ K, bias voltage $V_b = 300$ mV, set-point current $I_{sp} = 400$ pA, scale bar, 5 nm). Note the small number of defects. The total height scale is 200 pm. The zoomed inset (scale bar, 1 nm) shows the orientation of the crystallographic axes, clearly indicating B termination. (**c**) Spatially ($2 \times 2\,nm^2$) averaged STS on part of the surface displayed in the inset of **b**. Several features can clearly be distinguished within $\pm 20$ mV, which are marked as (i) to (v) and discussed in the text. Yellow to grey background in **c** indicates the energy range within which the gap opens, while the patterned area marks the region for potential in-gap states. $V_b = 50$ mV, $I_{sp} = 125$ pA, modulation voltage $V_{mod} = 0.2$ mV.

range. Around the same energy, the opening of a gap has been widely observed by a number of spectroscopic tools as mentioned above[16–24], including STS (refs 36–38). On the basis of the band structure displayed in Fig. 1b, the kinks marked by (i) can be ascribed to the Kondo hybridization between the $f$-band and the conduction band, which results in a decreased conduction electron density inside the hybridization gap below the Kondo temperature $T_K$ (ref. 40). Here, $T_K$ marks the crossover from (single ion) local moment behaviour at high temperature to entangled behaviour between $f$ and conduction electrons[41].

More importantly, we were able to disentangle several anomalies, which were hitherto not resolved individually by STS at higher temperature[36–38]. Benefitting from this improvement, we can investigate the fine structure of bulk/surface bands and go beyond a simple Kondo hybridization analysis, which is based on only one $f$-band and one conduction band[14]. Around $-13.5$ meV, there is a small peak marked by (ii). Excitations with similar energy have been reported before, for example, by ARPES ($-15$ meV) (ref. 15), X-ray photoelectron spectroscopy ($-15$ meV) (ref. 14) and inelastic neutron scattering (14 meV) (refs 42,43), yet with differing explanations as to its origin. As discussed below, this small peak is most likely related to the indirect tunnelling into the localized $\Gamma_8^{(2)}$ states. Compared with delocalized $f$-states, such localized $f$-states may give rise to only small anomalies in spectroscopy measurements[44].

Compared with peak (ii), peak (iii) (at around $-6.5$ meV) is very sharp and pronounced. Such a peak has been observed on different types of surfaces, including reconstructed ones[36–38], which clearly indicates that there are significant bulk contributions to this state. Very likely, the weakly dispersive structure of the hybridized $\Gamma_8^{(1)}$ band around the $X$-point along with the Fano effect can induce a peak in the conductance spectra at this energy level. In a Kondo system, the Fano effect is due to a quantum mechanical interference of electrons tunnelling into the localized states and the conduction bands[45,46]. Either a sharp drop (like feature (i)) or a pronounced peak will show up around the gap edge, depending on the tunnelling ratio between the two channels, as well as the particle-hole asymmetry of the conduction band. However, as has been reported previously, the spectrum deviates from a simple Fano model at low-temperature[36,38], indicating additional components to peak (iii) (see also discussion below). This is consistent with our inference

that the hybridized $\Gamma_8^{(1)}$ band resides within the big gap $\Delta_1$ and also contributes to the intensity of this peak. Hence, the position of peak (iii) can provide an indication with respect to the energy level of the $\Gamma_8^{(1)}$ band and therefore the size of the small gap $\Delta_2$. Note that its energy level is also comparable with the size of the small bulk gap observed by transport measurements[25–27]. Therefore, peak (i) to (iii) can directly be compared with the band structure in Fig. 1b. To verify the bulk/surface origins of these peaks at low-temperature, impurity, magnetic-field, and temperature dependences of STS have been conducted. As we will show below, besides bulk components, peak (iii) also contains components from the surface layer below 7 K.

Crucially, we also observe small anomalies (iv) and (v) at $\pm 3$ meV, which reside just inside the bulk gap $\Delta_2$ ( cf. also results on temperature-dependent STS spectra below). The shoulder-like shape of these small anomalies indicates the existence of two weakly dispersive bands or localized states near $E_F$. It is noted that both features at about $\pm 3$ meV also reveal spatial inhomogeneity (Supplementary Fig. 2), which—given the electronic inhomogeneity of even atomically flat surfaces[39]—hints at the surface origin of these states.

**Spatial dependence of the STS spectra.** For STM measurements, one possible way to distinguish bulk and surface states is to carefully investigate the tunnelling spectra at/near impurities or other defects, because the surface states are more vulnerable to such defects. Therefore, $g(V)$ was measured across two impurity sites at 0.35 K, shown in Fig. 3a,b. The bigger impurity at #A with an apparent height of $\approx 160$ pm is probably located on top of the surface, while the smaller one at #E (apparent height $\approx 50$ pm) is likely incorporated into the crystal. According to Fig. 3c, the $g(V)$-curves are all very similar for positions #B to #F. Even at position #A, that is, on top of the big impurity, the spectrum exhibits similarities; in particular all anomalies (i)–(v) can be recognized. In addition, a new peak occurs at $-10$ meV, which may be assigned to an impurity bound state. In Fig. 3d, we plot the height of the peaks (ii) to (iv) at different positions. A combined analysis of Fig. 3c,d reveals spatial stability of peak (ii), being consistent with the expectation for bulk states as discussed above. On the other hand, peaks (iii) and (iv) are not as stable as peak (ii); their heights are suppressed by both the big and the small impurity, which implies that at this temperature both peaks contain contributions from the states pertained to the surface.

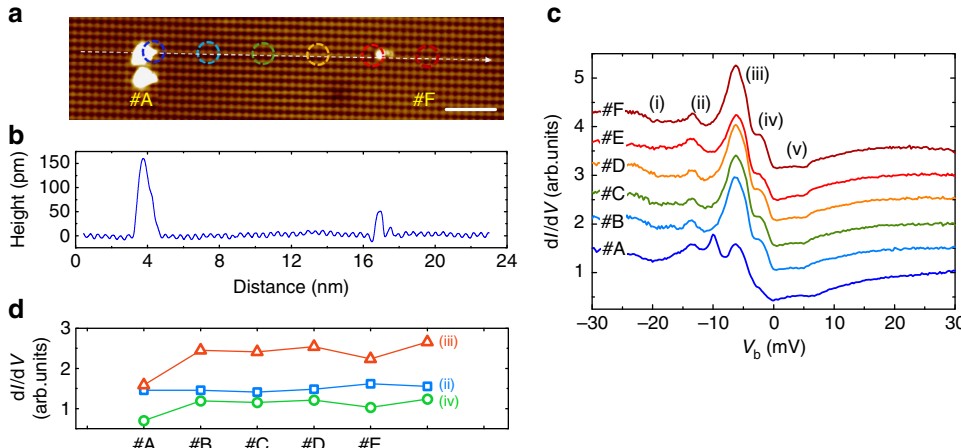

**Figure 3 | Spatial dependence of tunnelling spectroscopy. (a)** Topography of a non-reconstructed, B-terminated surface (scale bar, 2 nm) with two different types of defects, two large ones at position #A and a smaller one at #E. **(b)** Height scan along the dashed line indicated in **a**. **(c)** d$I$/d$V$-curves measured at six positions (denoted as #A to #F) equally spaced and marked by circles in **a**. Curves are offset for clarity. $T = 0.35$ K, $V_b = 30$ mV, $I_{sp} = 100$ pA, $V_{mod} = 0.3$ mV. **(d)** Maximum peak values of the differential conductance at $-13.5$, $-6.5$ and $-3$ mV obtained at positions #A to #F.

**Magnetic-field dependence of the STS spectra.** In Fig. 4a,b, $g(V)$-curves measured at sites #A and #C of Fig. 3a for different applied magnetic fields are presented. There is no distinct change detected up to the maximum field of 12 T for features (i) to (v), except an enhanced peak amplitude for the impurity state at $-10$ meV, see Fig. 4b. The magnetic-field independence of these states is consistent with the observation of metallic surface conductance up to 100 T by transport[26,47–49] and spectroscopic measurements[30,36]. This observation can be understood by considering a very small $g$-factor (0.1–0.2) of the $f$-electrons[50].

**Temperature dependence of the STS spectra.** We now turn to the temperature dependence of the features (i) to (v). The temperature evolution of the STS spectra was measured continuously on the same unreconstructed, B-terminated surfaces away from any defect between 0.35 and 20 K, see Fig. 4c. Above 15 K, the spectra show a typical asymmetric lineshape which arises from the Fano effect[45,46], being in good agreement with previous work[37]. Following the interpretation of ref. 45 the peak position in energy can be related to the gap edge, that is, the $\Gamma_8^{(1)}$ band in case of SmB$_6$, as discussed above. On cooling, the amplitude of peak (iii) increases sharply, accompanied by a sudden appearance of peaks (iv) and (v) below 7 K, with the latter effect being beyond thermal smearing (Supplementary Fig. 3). The low-temperature evolution of the spectra is clearly seen after the measured $g(V, T)$-curves were subtracted by the data at 20 K, see Fig. 4d. In an effort to quantitatively investigate the evolution of the spectra with temperature, we describe the low-temperature $g(V)$-curves by a superposition of four Gaussian peaks on top of a co-tunnelling model (Supplementary Fig. 4). However, fits to data obtained at higher temperature ($T > 10$ K) turned out to be less reliable (Supplementary Figs 5 and 6).

To further analyse the temperature evolution of peak (iii), we normalized the spectra by its size at $V_b = \pm 30$ mV. The resulting $g(T)$-values of peak (iii) are plotted in Fig. 4e. Clearly, a change in the temperature dependence is observed around 7 K. This is further supported by a comparison to data obtained by Yee *et al.*[36] (blue circles and blue dashed line) in a similar fashion but on a $(2 \times 1)$ reconstructed surface (which may explain the scaling factor, right axis). Also, the spectral weights of the $-10$ meV peak by Ruan *et al.*[38] (green squares) indicate a similar trend at $T \gtrsim 5$ K. Note that even the temperature evolution above about

7 K cannot be explained by a mere thermal broadening effect[36,38]. By tracing the temperature evolution of the d$I$/d$V$-spectra between about 7–50 K (refs 36–38) a characteristic energy scale of about 50 K was derived. This can be accounted for by the Kondo effect of the bulk states, with an additional contribution from a resonance mode[38], which is likely (as discussed above) related to the $\Gamma_8^{(1)}$ state. The same energy scale of $\gtrsim 50$ K has also been observed by transport[9,12] and other spectroscopic measurements[13,20,51,52]. However, below 7 K, the intensity of peak (iii) shows a sudden enhancement in Fig. 4e, indicating the emergence of an additional energy scale. Considering the fact that this new energy scale, as well as many other exotic transport phenomena related to the formation of a metallic surface[10–12] set in simultaneously, the increase in intensity of peak (iii) (as well as the appearance of peaks (iv) and (v)) is expected to rely on the same mechanism that is responsible for the formation of the metallic surface. Both observations appear to evolve out of the bulk phenomena associated with the primary hybridization gap at elevated temperatures. In the following section, we will argue that this new energy scale is related to the suppression of the Kondo effect at the surface.

## Discussion

In this study, the topographic capabilities of the STM allow us to distinguish features (i) to (v) on non-reconstructed (001) surfaces of a single termination and without apparent defects. Therefore, we can simply exclude the possibility that they are driven by surface reconstructions or defects. Especially the observation of new states on clean surfaces below about 7 K indicates that the exotic properties of SmB$_6$ are intrinsic rather than due to impurities. The observation of well-resolved features in our tunnelling spectra (discussed above) enables the direct comparison with results of bulk band structure calculations[4–7,18,53]. This not only reveals the energy levels of the multiplet $f$-states, but can also reconcile the long-standing debate of 'small' versus 'large' bulk gap in SmB$_6$ (ref. 14). Consequently, our data shows that a dedicated hybridization model with two—instead of one— multiplet $f$-states is necessary to interpret the low-energy properties of SmB$_6$. In particular, peak (iii) has multiple components including bulk and surface states, the ratio of which changes dramatically with temperature.

It is widely accepted that the electronic properties of SmB$_6$ can be divided into several temperature regions, which are based on

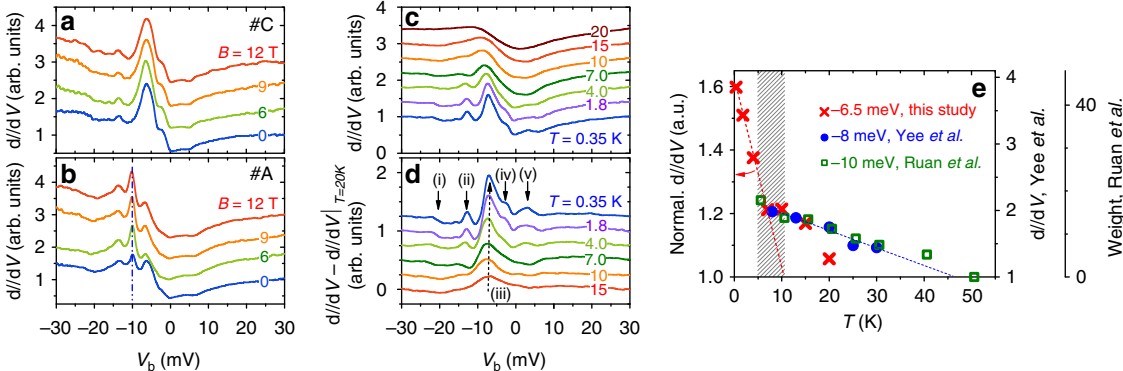

**Figure 4 | Magnetic-field and temperature dependence of STS spectra.** Tunnelling spectra measured at magnetic fields up to 12 T and 0.35 K on the top of (**a**) site #C away from any impurity and (**b**) site #A atop a small impurity. (**c**) Evolution of d$I$/d$V$-curves from 0.35 K to 20 K. To compare the data, a small linear background is subtracted from the raw data. (**d**) Difference tunnelling conductance after subtracting the $g(V)$-data measured at 20 K. (**e**) Temperature dependence of the intensity of peak (iii). Results for intensities by Yee et al.[36] (blue) and spectral weights by Ruan et al.[38] (green) are shown for comparison (right axes). Curves are offset for clarity in (**a**–**d**). $V_b = 30$ mV, $I_{sp} = 100$ pA, $V_{mod} = 0.3$ mV.

transport measurements[18,26], as well as other probes, like ARPES (refs 18,20). Apparently, 5–7 K is a crucial regime, where the temperature-dependent properties undergo significant changes. Above this range, the electronic states in SmB$_6$ are governed by the Kondo effect of the bulk[14,16,17]. At lower temperatures, several interesting observations—in addition to that of the saturated resistance—were made. For example, the Hall voltage becomes sample-thickness independent[11]; the angular-dependent magnetoresistance pattern changes from fourfold to twofold symmetry[26]; and the development of a heavy fermion surface state is found by magnetothermoelectric measurements[54]. These experimental facts provide convincing evidence for the formation of (heavy) surface states just around 5–7 K, which is in line with the appearance of a new energy scale.

Recently, a surface Kondo breakdown scenario was proposed based on the reduced screening of the local moments at the surface. As a result, the Kondo temperature of the outmost layer, $T_K^s$, can be strongly suppressed, resulting in a modified band structure[55]. Slab calculations further show that below $T_K^s$ surface $f$-electrons gradually hybridize with conduction electrons at the surface and form a weakly dispersive band close to $E_F$ (refs 50,53). Remarkably, very narrow peaks with strongly temperature-dependent STS spectra near $E_F$ are regarded as a smoking gun evidence for a surface Kondo breakdown scenario[53]. On the basis of our experimental results, $T_K^s$ is inferred to be around 7 K, being about an order of magnitude smaller than $T_K$. The evolution of our tunnelling spectra below about 7 K also fit excellently to the theoretical prediction and the related calculations for STS. In such a scenario, the additional component at $-6.5$ meV and shoulders at $\pm 3$ meV are related to the heavy quasiparticle surface states, the formation of which supplies an additional tunnelling channel in particular into the $f$-states. This provides a highly possible origin for the metallic surface states and a reasonable explanation to the various experimental observations listed above.

We note that theoretically a surface Kondo breakdown effect does not change the topological invariance of SmB$_6$, which is determined by the topology of the bulk wave functions. Therefore, the surface-derived heavy quasiparticle states could still be topologically protected. Experimentally, for such topologically protected surface states backscattering is forbidden in quasiparticle interference (QPI) patterns as measured by STM (ref. 56). In line with this prediction and as shown in the Supplementary Fig. 7, no clear quasiparticle interference pattern could be detected so far from our results, which is similar to the observation by Ruan et al.[38].

## Methods

**Sample preparation and STM measurements.** All samples were grown by the Al-flux method. A cryogenic (base temperature $T \approx 0.35$ K) STM with magnetic-field capability of $\mu_0 H \leq 12$ T was utilized. Three SmB$_6$ single crystals were cleaved a total of five times *in situ* at $\approx 20$ K to expose a (001) surface. Cleaved surfaces were constantly kept in ultra-high vacuum, $p < 3 \times 10^{-9}$ Pa. Tunnelling was conducted using tungsten tips[57], and the differential conductance ($g(V)$-curve) is acquired by the standard lock-in technique with a small modulation voltage $V_{mod} \leq 0.3$ mV. On our best cleaved sample, the size of non-reconstructed surface areas can reach up to $100 \times 100$ nm$^2$.

**Analysis of STS spectra.** In principle, the low-temperature $g(V)$-curves can be well described by a superposition of four Gaussian peaks on top of a Fano model (see example of $g(V, T = 0.35$ K) in Supplementary Fig. 4) or more elaborate hybridization models[45,46] (Supplementary Fig. 6). A similar procedure with only one Gaussian was employed in ref. 38. However, fits are less reliable at elevated temperature. Instead, our spectra measured at different $T$ in zero field overlap nicely for $V_b < -25$ mV and $V_b > 10$ mV, such that they can be normalized by using very similar factors. Consequently, we can directly trace the temperature dependence of the peak height (at least for peak (iii)) by measuring the normalized peak intensity as shown in Fig. 4e. Note that the intensities of peak (iii) as obtained from Fig. 4d, that is, after subtracting the 20 K-data, yield very similar values as those shown in Fig. 4e from normalized spectra.

**Data availability.** The data supporting the findings of this study are included within this article (and its Supplementary Information files), or available from the authors.

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

## Acknowledgements

We acknowledge valuable discussion with J. W. Allen, P. Coleman, X. Dai, I. Eremin, O. Erten, C.-L. Huang, Deepa Kasinathan, B. I. Min, C. J. Kang, G. Sawatzky, Q. Si and P. Thalmeier. This work was supported by the Deutsche Forschungsgemeinschaft through SPP 1666 and by the Defense Advanced Research Agency (DARPA) under agreement number FA8650-13-1-7374. L. J. acknowledges support by the Alexander-von-Humboldt foundation.

## Author contributions

S.W., L.H.T. and F.S. designed the research. L.J. and S.R. conducted the STM experiments. D.J.K. and Z.F. provided the samples. All authors contributed to the discussions and the manuscript.
