## [Peer Review File · Nature Communications]

Reviewers' comments:

Reviewer #1 (Remarks to the Author):

This paper reports the discovery of several in-gap states in SmB₆ using STM at low temperatures down to 0.35 K with external magnetic fields up to 12 T. With a high energy resolution of about 0.5 meV, the authors declare the observation of a surface Kondo effect, where the surface states hybridize with one of the f-states, opening up a small hybridization gap at much lower Kondo temperature (T_K), which may explain several experimental results well.

The SmB₆ Kondo insulator system has been one of the most puzzling materials due to its peculiar transport properties and the possibility of being a topological insulator with strong interactions. The STM results reported here have very high quality, and cover the temperature and magnetic field ranges that are crucial but were not available before. The results will shed important new light on the current debate on the SmB₆ system. Therefore, I recommend the publication of the paper in Nature Communications. Below I list a few minor points that the authors should address during the revision.

1. The authors have obtained high quality Boron-terminated surface, which is free of defects and surface reconstructions, allowing them to investigate intrinsic local electronic structures. Because the surface states play an important role in the key physics of SmB₆, no matter what the origin is (topologically trivial/nontrivial), quasiparticle interference (QPI) signals should be detectable at such clean surfaces. The dispersion of QPI could shed important light on the surface electronic structure of this material. However, the authors did not mention this point at all. I think most readers working this field would be curious about it.

2. The observed -6.5 meV peak is referred to those in-gap peaks found in previous STS results at higher temperatures [39-41], which is supported by the similar energy scale and temperature dependence (Fig. 2c). This peak is claimed by the authors to contain contributions from some localized states and also surface states below 7 K. However, there is one statement in Page2 "This peak then contains a component of the conduction band due to a quantum mechanical interference of electrons tunneling into the localized states and the conduction bands (referred to as Fano resonance or co-tunneling)", which is not accurate. The main point of ref. [41] is that the peak in STS cannot be directly reproduced by co-tunneling, and has a different onset temperature ($\sim 40-50$ K) from the bulk Kondo temperature (should be above 100 K). The finding here of a further increase of the peak below 7 K may provide essential information to uncover the nature of this peak, and in this sense the results are consistent with findings in [41].

3. Although the proposed narrow surface Kondo gap ~ 3 meV formed below 7 K provides good explanations for some experimental facts such as quantum oscillations, it seems inconsistent with the low temperature resistivity saturation below 5 K, where a gap opening in principle should lead to a rise in resistivity. The authors should discuss the characteristic R-T behaviour in the presence of such a gap.

Reviewer #2 (Remarks to the Author):

The submitted manuscript addresses the evolution of spectral features, as measured by STM, in the putative strongly correlated topological insulator compound, SmB₆. Unresolved issues including the topological or trivial nature of the surface states, the mechanism for the opening of the bulk gap, the role of Kondo physics, and the potential for new topological orders in strongly correlated systems

makes this material still an immensely interesting one to study.

The work by Jiao et al. aims to address some of the outstanding issues, with particular attention to identifying the role of multiplet f-states, possibly the transition of the electronic structure from 3D to 2D at a newly identified 'energy scale', and the overall evolution of the density of states spectrum, as determined by STM measurements. Their data are unique because they are at the lowest temperatures reported by STM on this compound, they include information about tuning with magnetic field, and very importantly that they are on pristine unreconstructed surface from which to draw inferences.

However, the article suffers severely from being unfocussed and excessive over interpretation of all of the features in their spectral curves. This happens to the point that there are inconsistencies. For example, it is claimed that the loss of coupling to f-electrons leads to a peak in the dI/dV curves at one energy whereas they effectively say that another peak arises because of a coupling to f-electrons. Effectively, the same form of double interpretation is used in analysis and identification of which part of the spectral curve comes from a tunneling interference effect leading to Fano line shapes and which parts come from f-states. As a third example, similar inconsistent arguments are used on defects to attribute certain features to the bulk, some impurity bound states, etc. In short, there is no real systematic study to truly confirm any of these assertions.

The new energy scale, described in this paper, and identified by the rapid rise in the largest peak in dI/dV curves is an interesting observation but is essentially hand waved away as a 3D-2D transition based on ARPES experiments. At the very least, the authors could explain why such a transition would lead to this peak amplitude increase. There are certainly simple explanations having nothing to do with exotic physics. On the other hand, if the 3D-2D transition is in fact key to understanding this compound the discussion in this paper does nothing to aid that. I appreciate the many ideas that are expressed in the paper but none are treated seriously in the context of the observations.

In summary, I cannot recommend the publication of this manuscript in Nature Communications. While the observations and access to data on an unreconstructed surface and at low temperatures are important, there is insufficient detailed study of many of the claims and assertions to warrant publication.

REVIEWERS' COMMENTS:

Reviewer #1 (Remarks to the Author):

In the revised manuscript, the authors have clarified their statements on the origin of the -6.5 meV peak and the additional conducting channels below the surface Kondo temperature. The added Figure 1 makes the physical picture much clearer. They have also included detailed QPI results in the supplementary materials, although it is quite unfortunate that they are inconclusive. The observed novel energy scale at low temperature has been shown to be related to the surface Kondo effect experimentally through STS evolutions.

In conclusion, I think the manuscript is significantly improved by the revision, and it deserves publication in Nature Communications.

Reviewer #2 (Remarks to the Author):

I thank the authors for their extensive revision of the manuscript. It is much more clear and direct in its assertions and analysis making for a more compelling paper. With the unique experimental data and the fair analysis at this point I am more in favor of publication but would like to see the following points addressed:

1) In places throughout the paper T_K scale and T^* scale for heavy fermion coherence are interchanged and this is not correct. Unfortunately, it has become more common for this mistake to happen but these two temperature scales can be well separated and the existence of the former doesn't necessitate the latter. I would recommend sharpening the use of the terminology.

2) In attributing some changes in the spectra on impurity #A to presence of surface states which are present at low T , the follow-up experiment to validate this would be to check that the spectra are restored at higher temperature. This would wrap up the loose end of this part of the analysis.

3) The use of standard Fano analysis (as seen in SI Fig 4) on the background spectrum is puzzling: since the large peak (peak iii), for higher T , is described as corresponding to the f-level taking part in hybridization and forming a hybridization gap how does it also participate in the tunneling interference effect to generate the background Fano lineshape? Or does that Fano lineshape come from a different source?

In Maltseva and Coleman, for example, the high peak would be associated with the flattening out of the valence band (e.g. peak iii). Is that the way the authors wish to represent their spectra here? And if so, does the enhancement of peak iii) at $T < 7K$ mean that the onset of the new heavy fermion hybridization or Kondo screening is directly adding to this same f-level hybridization?

Response to the first referee:

We highly appreciate the first referee for recommending our manuscript for publication in Nature Communications. We are highly encouraged by the referee's appreciation of "*The STM results reported here have very high quality, and cover the temperature and magnetic field ranges that are crucial but were not available before. The results will shed important new light on the current debate on the SmB₆ system.*"

Moreover, we thank the referee for pointing out three issues; we are confident that the manuscript clearly gained by addressing these points.

1. The authors have obtained high quality Boron-terminated surface, which is free of defects and surface reconstructions, allowing them to investigate intrinsic local electronic structures. Because the surface states play an important role in the key physics of SmB₆, no matter what the origin is (topologically trivial/nontrivial), quasiparticle interference (QPI) signals should be detectable at such clean surfaces. The dispersion of QPI could shed important light on the surface electronic structure of this material. However, the authors did not mention this point at all. I think most readers working this field would be curious about it.

We fully agree with the referee that QPI data is important and highly desired. Indeed, we have spent quite some time and efforts to obtain high-quality QPI results. We now provide selected dI/dV-maps and their Fourier transform at five bias voltages in the Supplementary Fig. 7. In the dI/dV-map, the lattice structure as well as a small number of impurities are well resolved. The Bragg points of the lattice are also clearly resolved in the Fourier transform. However, any signature of a clear QPI pattern is still missing within ± 10 meV. We believe that this is a result of a forbidden scattering due to the opposite spin of states at \mathbf{k} and $-\mathbf{k}$ on the Dirac cone. However, we have no clear evidence for such a claim as there are other possible reasons for the non-observance of QPI patterns. Hence, we include the QPI data in the Supplementary information (Fig. S7) of our manuscript along with the above remarks (last paragraph of the discussion) for the readers' information but do not wish to derive any claim from it.

2. The observed -6.5 meV peak is referred to those in-gap peaks found in previous STS results at higher temperatures [39-41], which is supported by the similar energy scale and temperature dependence (Fig. 2c). This peak is claimed by the authors to contain contributions from some localized states and also surface states below 7 K. However, there is one statement in Page2 "This peak then contains a component of the conduction band due to a quantum mechanical interference of electrons tunneling into the localized states and the conduction bands (referred to as Fano resonance or co-tunneling)", which is not accurate. The main point of ref. [41] is that the peak in STS cannot be directly reproduced by co-tunneling, and has a different onset temperature (~ 40 -50 K) from the bulk Kondo temperature (should be above 100 K). The finding here of a further increase of the peak below 7 K may provide essential information to uncover the nature of this peak, and in this sense the results are consistent with findings in [41].

We thank the referee for pointing out this inaccuracy. We agree with the referee that peak iii) has an additional component other than the one originating from the co-tunneling effect, especially at low temperatures. We have changed the corresponding sentence (last line of p.2 – first two lines of p.3) and now state explicitly that peak iii) contains additional components beyond co-tunneling. To make this point very clear we also added a corresponding statement on p.5, left column, line 8 – 12, in which the potential reader is again reminded of this additional contribution.

3. Although the proposed narrow surface Kondo gap ~ 3 meV formed below 7 K provides good explanations for some experimental facts such as quantum oscillations, it seems inconsistent with the low temperature resistivity saturation below 5 K, where a gap opening in principle should lead to a rise in resistivity. The authors should discuss the characteristic R-T behaviour in the presence of such a gap.

The exponential increase of the temperature dependent resistance of SmB₆ is well understood in

terms of the Kondo effect of the bulk. In case of SmB_6 a gap of the order of 20 meV emerges upon cooling to below about 100 K putting this material in the class of the so-called Kondo insulators. However, at the surface the interactions between the $4f$ states are modified due to the lattice symmetry resulting in a much smaller surface Kondo temperature. Upon decreasing the temperature below this surface- T_K , heavy quasiparticles gradually form on the surface layer. At the surface, the energy levels of the quasiparticle bands are therefore much smaller (likely $\sim \pm 3$ meV, see Fig. 4d) compared to the bulk states and hence, located inside the latter. In our line of argument, this is manifested by the appearance of peaks iv) and v). For peak iii), the sharp increase of its intensity below 7 K arises from a contribution due to the Kondo coupling between conduction electrons and the multiplet f -states at the surface or, possibly [52], the f -states in the second layer, which is indistinguishable experimentally. These surface quasiparticles can increase the LDOS around E_F relative to the expected evolution of the (bigger) bulk gap (for comparison see Fig. 3C of Ref. 37 and Fig. R1 below) and provide additional conducting channels at the surface. This can be seen from Figs. 4(c) and (d) where the LDOS very close to E_F remains more or less constant at low temperatures. We note that even though the peaks at ± 3 meV are ascribed to signatures of heavy quasiparticle surface bands, it does not have to form a well-defined or fully opened gap on the surface, given the very low energy scale of the Kondo effect on the surface.

Fig. R1: Evolution of dI/dV -curves from 0.35 K to 20 K without shift. Data are normalized at -30 mV. In order to compare the data, a small linear background is subtracted from the raw data. The inset shows the temperature dependent depth of the gap at zero bias voltage, which possesses a minimum around 7 K.

To clarify this point, we have modified the explanations and added to our discussion in the revised manuscript. In particular, the third paragraph of the discussion (“Recently, a surface Kondo breakdown ...”) has been modified considerably. Two sentences were added at its end directly addressing the referee’s comment.

In response to the second referee:

The work by Jiao et al. aims to address some of the outstanding issues, with particular attention to identifying the role of multiplet f -states, possibly the transition of the electronic structure from 3D to 2D at a newly identified 'energy scale', and the overall evolution of the density of states spectrum, as determined by STM measurements. Their data are unique because they are at the lowest temperatures reported by STM on this compound, they include information about tuning with magnetic field, and very importantly that they are on pristine unreconstructed surface from which to draw inferences.

We thank the referee for the comments and for appreciating the high quality and importance of our work. Our aim was indeed to elucidate on the low temperature properties of SmB₆, which are complicated by the puzzling number of effects named by the referee. By our work, many of these issues can be understood in a unified and consistent picture provided in the manuscript.

However, the article suffers severely from being unfocussed and excessive over interpretation of all of the features in their spectral curves. This happens to the point that there are inconsistencies. For example, it is claimed that the loss of coupling to f -electrons leads to a peak in the dI/dV curves at one energy whereas they effectively say that another peak arises because of a coupling to f -electrons.

In the revised version, we have taken these very serious criticisms into consideration and made major modifications to the manuscript in order to present our findings more clearly and consistently. This is prominently manifested by the fact that we now include an improved and more detailed sketch (as well as its discussion) of the band structure of SmB₆ in the introduction. Our hope is that by doing so it is immediately obvious which states may contribute to the density of states (particularly if they are hybridized) and in which case the Kondo coupling results in a gap formation which, in consequence, diminishes the LDOS. The latter is very likely the case for anomalies i) below about 20 meV for the bulk, and possibly for iv) and v) for energies smaller than about 3 meV for the surface Kondo effect. Note that although the fine structure of the $j = 5/2$ state of the f -electrons in SmB₆ is not well resolved by experiment yet, theoretically, a common feature is shared among several calculations based on different methods as we addressed in the revised introduction (first paragraph of the introduction). We also provided a simple simulation of the band structure in Supplementary Fig. 1, which is comparable with the band structure calculations and can capture the main feature of the Kondo gap opening around E_F of the bulk. These facts indicate the applicability of our sketched band structure.

The referee's statement was well justified for our original discussion of peak ii) which we assign to the $\Gamma_8^{(2)}$ band that cannot hybridize with the conducting band. Peak-like features in the LDOS arise naturally from the nearly flat bands or local resonance states. The fact that the $\Gamma_8^{(2)}$ multiplet does not hybridize with the conduction band is due to its symmetry (indicating an incoherent/localized f -state) and results in an only rather weak/small peak-like anomaly. In other words, the hybridization is an important component in determining the amplitude of a peak. An analogy can be found in ARPES result obtained on CeB₆ and SmB₆ at 38 K: the localized f -states in CeB₆ are manifested by small broad peaks, while sharp strong peaks are detected in SmB₆ due to hybridized f -bands [43]. We have considerably modified the corresponding paragraph (page 2, center of right column) to explain these effects.

We wish to point out that in case of tunneling into a Kondo material (heavy fermion metal or Kondo insulator) by means of STM, there are two channels available, the quasiparticle as well as the conduction electron channel, which can interfere with each other. This can be described by a Fano model or other, more sophisticated co-tunneling models [44, 45]. In this case, a reduction or/and an increase in the STS spectrum near the Fermi level can arise depending on the value of q or t_f/t_c , the ratio of the tunneling probabilities into the quasiparticle states and the conduction band as well as on the particle-hole asymmetry. This is shown in the simulation below and also in previous works [36 - 38]. Note that these models and simulations are based on the simplified case in which only one f -band hybridizes with one conduction band. However, as clearly stated in our main text and Supplementary Figures there are likely several components with different temperature dependences. For this reason, a direct and quantitative peak analysis of peak iii) with its different contributions versus temperature is hampered.

Fig. R2: Simulated dI/dV -curves as a function of q or t/t_c based on (a), (c) the Fano model and (b), (d) the model by Maltseva *et al.* [44] model, respectively. The models and the parameters used for these simulations are similar to those found in Ref. [37]. A hump-dip-like feature can be obtained by changing q or t/t_c . The hybridization gap is assumed to be 20 meV.

Effectively, the same form of double interpretation is used in analysis and identification of which part of the spectral curve comes from a tunneling interference effect leading to Fano lineshapes and which parts come from f -states.

Obviously, we haven't been clear here either. Yet, there is a number of effects coming into play here again. There are hybridized and non-hybridized f -states in the bulk and – at lower temperatures – also at the surface which are to be distinguished. The interference effects observed in tunneling spectroscopy and responsible for the Fano lineshape are brought about by the existence of two channels (see above) one of which is made up by the quasiparticles and hence, involves some f -states. However, also the weakly hybridized f -band $\Gamma_8^{(2)}$ can contribute as STS still measures the DOS. For peak iii) at -6.5 meV we infer at least two components at **high** temperature: First, the Fano (or co-tunneling) effect of the composite heavy quasiparticle band and second, the weakly dispersive $\Gamma_8^{(1)}$ -band itself. Both components are of **bulk** origin. Note that in the case of two f -bands hybridizing with one conduction band, like in SmB_6 , one band ($\Gamma_8^{(1)}$) shows weak dispersion, while the other (Γ_7 -band) opens a well-defined gap. A simplified simulation of the hybridization gap opening process is presented in the Supplementary Fig.1 in the revised version. However, the ratio of the two contributions to peak iii) cannot easily be disentangled, as pointed out above and in the manuscript (sentences at end of p.2 – beginning of p.3 and p.5, end of first paragraph of the discussion).

Only at **low** enough temperatures (below 7 K) will the surface Kondo effect give rise to a sufficiently strong coupling between the conducting electrons and the f -electrons on the **surface** layer. This surface Kondo effect will delocalize f -electrons in the surface layer, resulting in a dramatic increase of the peak amplitude around -6.5 mV and the appearance of shoulders at ± 3 mV. Based on this assumption, a DMFT calculation of STS spectra has been conducted by Peters *et al.* [52], which indicate the emergence of a sharp peak-like feature with strong temperature dependence around E_F , just as observed by us.

We are positive that the revised Figure 1 helps in clarifying all these issues. In addition, we have made numerous respective changes throughout the manuscript, but particularly to the discussion.

As a third example, similar inconsistent arguments are used on defects to attribute certain features to the bulk, some impurity bound states, etc. In short, there is no real systematic study to truly confirm any of their assertions.

One of the main advantages of STM is its capability to combine topography and spectroscopy, i.e. spectroscopy can be conducted at well-defined locations on the surface. One possible way to distinguish surface states and bulk states is to make use of impurities at the surface, and investigate the spectroscopic response to such impurities. In our case, we find an anomaly in the dI/dV -data at around -10 mV, which exclusively appears on top of a particular type of impurity; hence, it is very reasonable to interpret this anomaly as an impurity state. We note that this assignment is completely independent of the bulk-vs-surface issue.

In an effort to distinguish between bulk and surface contributions to certain spectral features we investigated their response to the proximity to impurities. Due to the broken translational symmetry of the top layers, surface states form. Therefore, surface states are typically more sensitive to impurities on the topmost layer than bulk states. It is obvious from Fig. 3(d) peaks iii) and iv) are partially suppressed by impurities, indicating contributions from surface states. On the other hand, peak ii) is pretty robust against the impurity and likely corresponds to a bulk state.

It is the combination of impurity, magnetic field and temperature dependence of the peak amplitudes, along with a comparison to the bulk band structure, which let us conclude that peak ii) is dominated by bulk states, peak iii) has both bulk and surface contribution, and peak iv) is dominated by surface states at the lowest temperature. We note that the surface states mentioned here are not necessarily topologically protected.

We realized that these discussions have been distributed throughout several paragraphs of the original manuscript, which may be not a very straightforward way of presenting our results to the readers. In the revised version, we have merged this information in consecutive sub-chapters (starting on p.3 – left column of p.4: Spatial dependence, Magnetic field dependence, temperature dependence) and (hopefully) improved on the presentation. We hope these discussions now provide the required clarity.

The new energy scale, described in this paper, and identified by the rapid rise in the largest peak in dI/dV curves is an interesting observation but is essentially hand waved away as a 3D-2D transition based on ARPES experiments. At the very least, the authors could explain why such a transition would lead to this peak amplitude increase. There are certainly simple explanations having nothing to do with exotic physics. On the other hand, if the 3D-2D transition is in fact key to understanding this compound the discussion in this paper does nothing to aid that.

The referee is certainly correct: The new energy scale is a key finding and as such we have to introduce and explain it well. However, we wish to point out that the 3D to 2D transition is not simply based on ARPES work only. In fact, there is quite a number of papers reporting the surface (2D) conductivity just below ~ 5 K by using different techniques, the most convincing results stemming from resistance and Hall effect measurements [10-12,26]. The reported appearance of surface states in SmB_6 in general, and the formation of metallic surface states found by transport measurements in particular, occur in the same temperature range as the steep increase of the amplitude of peak iii) and therefore, it is natural to ascribe the surface states as an indication for 2D conductance. To further support our conclusion we show in the inset of Fig. R1 of this response letter the temperature dependence of the dI/dV -values at E_F ; we interpret its increase below 7 K as an increase of the conduction electron DOS (not to be confused with the f -states) due to Kondo coupling at the surface. We note that by citing the above references we want to compare with experimental reports instead of jumping to conclusions about a topological nature of these surface states frequently found among the literature.

For clarity, we have removed the argument of 3D to 2D transition from the revised discussion part. As for the increasing amplitude of peak iii) we agree with the referee that only a very brief introduction to the corresponding scenario was provided. We have now elaborated on this in the revised discussion part (page 5, right column) interpreting its origin and the STS spectrum through a surface Kondo breakdown effect [49,52,54]. The relationship between the surface Kondo effect and the rapid rise of peak iii) in the dI/dV -curves can be understood in terms of a Kondo lattice model: above T_K^s (the Kondo temperature at the surface) the f -electrons are localized in the surface layer. Below T_K^s , the f -electrons hybridize with the conduction electrons which gives rise

to enhanced tunneling. This effect has been calculated in Refs. [44, 49, 52], which fit well to our observation. (Please, note also our related response to the third question by the first referee).

There are certainly simple explanations having nothing to do with exotic physics.

To our best knowledge, there are only a few other effects which were discussed to possibly induce a peak close to the Fermi energy in SmB_6 ; it could be due to bulk states [16], trivial polarity-driven surface states [23], Rashba-split surface states [24], a valence change or structural relaxation in the surface layer [5]. However, ARPES measurements are typically not conducted down to low enough temperatures (with the exception of [20, 24]). In particular, the results reported in Ref. [16] were obtained at 38 K, which is well above the new energy scale (~ 7 K), and they are actually consistent with our interpretation that above 7 K the bulk states gradually dominate with increasing temperature. The other two scenarios seem to be not supported by other ARPES studies [17-22]. One more-than-likely reason for the difficulties in any large-scale, area-averaged spectroscopy (including ARPES) is the absence of large, non-reconstructed (100) surfaces, i.e. there is likely an integration over different types of surface termination (as pointed out in the manuscript on p.2, left column). Here, the atomic resolution of our STM/S measurements is a clear advantage. Nevertheless, these scenarios cannot explain the sudden rise of the -6.5 mV peak below 7 K. The same holds for the proposed valence change. Structural relaxation effects have been calculated by Kim *et al.*, [5], but they are expected to induce pronounced surface states at much higher temperature than 5-7 K. In view of the observed temperature dependence, the surface Kondo breakdown effect, which is simply due to the reduced screening of local moments at the surface, provides the only reasonable explanation. Moreover, we do not consider this to be exotic since a reduced screening at the surface is to be expected.

I appreciate the many ideas that are expressed in the paper but none are treated seriously in the context of the observations.

Again, we thank the referee for pointing out some shortcomings in our manuscript and acknowledging our ideas. In consequence, we have thoroughly restructured our manuscript. Also, the discussion of the bulk and surface Kondo effect, which elucidates on the origin and the consequences of the new energy scale, has been improved (p.2 right column; p.5 upper left paragraph; p.5 right center). We emphasize that our experimental findings may reconcile many discrepancies in SmB_6 . Theoretically, our conclusions are well supported by many band structure calculations including the most sophisticated slab calculations using DMFT [52].

Response to the referee comments

We are delighted by the very positive assessment of the referees with respect to our revisions. Once more we wish to thank the referees for their very helpful comments which helped us to put forward these revisions. In the following we address the remarks of reviewer #2 in response to our revised version of the manuscript point by point.

1) In places throughout the paper T_K scale and T^ scale for heavy fermion coherence are interchanged and this is not correct. Unfortunately, it has become more common for this mistake to happen but these two temperature scales can be well separated and the existence of the former doesn't necessitate the latter. I would recommend sharpening the use of the terminology.*

Here, the referee touches on an indeed intriguing, and controversially discussed, issue. The T_K scale in a strict sense only refers to the single-ion case whereas the T^* scale is used to describe the Kondo lattice (here we hope to interpret the referee properly). In our manuscript we have adopted a view in which the T_K scale as defined for the single-ion case is simplifyingly “carried over” to the Kondo lattice case, as discussed, e.g. in our new Ref. 41 (it is defined in chapter 1.2 of Ref. 41 and then used in the Kondo lattice description, chapter 1.3 and the Doniach phase diagram). In fact, it was argued (Y. Yang *et al.*, Nature **454**, 611 (2008)) that T_K and T^* both derive from the single-ion coupling to the conduction electrons, with T^* being determined by the near-neighbor RKKY interaction.

However, the situation in SmB_6 is complex. The intermediate valence of Sm implies that there can be valence fluctuations and, importantly, the widths of the crystal field levels may well be of the order of the crystal field splitting, further complicating coherence considerations. To the best of our knowledge coherence issues in SmB_6 are not well investigated. In our view, one approach to follow up on these questions in this material was to study various dilutions of SmB_6 (see e.g., M. Kasaya *et al.*, Solid State Comm. **33**, 1005 (1980)); such studies are under way.

In order to comply with the referee’s request for sharpened terminology we have added a sentence (end of second paragraph of the Results section: “Here, T_K marks ...”). This sentence, which follows the first use of T_K in our manuscript, describes how we wish to use T_K . We also wish to point out that T_K does not appear otherwise throughout the manuscript except in the second-to-last paragraph of the Discussion section where it is compared to the surface Kondo temperature. In order to avoid confusion in this paragraph (discussing the surface Kondo effect) we modified the third sentence “Slab calculations further show ...” by replacing the former phrase “... f electrons gradually become coherent ...” with “... surface f -electrons gradually hybridize with conduction electrons at the surface ...”. We hope this modified sentence not only clarifies the usage of T_K^S but also makes the relation between T_K and T_K^S more clear.

2) In attributing some changes in the spectra on impurity #A to presence of surface states which are present at a low T , the follow-up experiment to validate this would be to check that the spectra are restored at higher temperature. This would wrap up the loose end of this part of the analysis.

Again, the referee points out an important issue. At the time the experiments reported here were carried out we did not pay attention to this bigger impurity because we were unable to determine the exact nature of this impurity (in particular, non-magnetic versus magnetic). Therefore, we did not attempt to investigate the temperature dependence of the spectra at these impurities. We should note here that it requires a time-consuming procedure to track a particular sample position upon changing the temperature in our STM, which is quite an effort.

However, we realized later on that the question of non-magnetic versus magnetic impurity may actually hold the key for resolving the issue of the presence of *topologically* protected surface states. We therefore have recently focused on the investigation of impurities in doped samples (some of the samples we now investigate were discussed in Kim *et al.*, Nat. Mat. **13**, 466 (2014)). Our studies on impurities (non-magnetic and magnetic) in these samples support the conclusions drawn in the present manuscript. Yet, we consider these as “follow-up experiments” (as the referee puts it), in particular the discussion of non-magnetic versus magnetic impurities.

3) The use of standard Fano analysis (as seen in SI Fig 4) on the background spectrum is puzzling: since the large peak (peak iii), for higher T, is described as corresponding to the f-level taking part in hybridization and form a hybridization gap how does it also participate in the tunneling interference effect to generate the background Fano lineshape? Or does that Fano lineshape come from a different source?

In Maltseva and Coleman, for example, the high peak would be associated with the flattening out of the valence band (e.g. peak iii)). Is that the way the authors wish to represent their spectra here? And if so, does the enhancement of peak iii) at $T < 7K$ mean that the onset of the new heavy fermion hybridization or Kondo screening is directly adding to this same f-level hybridization?

An analysis of tunneling spectra (obtained by STM or the point contact method) using standard Fano model is used quite often (e.g. Refs. 13, 36-38) to derive the parameters of SmB₆. Our Supplementary Figs. 4-6 display our efforts to follow this procedure. However, as stated in our manuscript, “fits to data obtained at higher temperature ($T > 10$ K) turned out to be less reliable ...”. It is important to note that “a dedicated hybridization model with two—instead of the typically used one—multiplet *f*-states is necessary to interpret the low-energy properties of SmB₆.” The applicability of a simplified analysis considering only one band in some cases may be due to a reduced resolution by thermal smearing effects or the surface conditions (note that in an effort to avoid these issues we went to $T = 0.3$ K and well characterized surfaces). At high temperatures, we assume that also the Γ_7 level is involved in the Fano effect and plays a role in the bulk property of SmB₆, but the tunneling ratio between the two channels cannot precisely be determined in this compound. To further answer the referee’s question, it is important to refer to our simulation in Fig. 1 and Supplementary Fig. 1: in the case of two nearly degenerate *f*-bands hybridizing with the *d*-band, the resulting (hybridized) $\Gamma_8^{(1)}$ band is rather flat and quite different from the one *f*-band case. At low temperatures, we certainly think the referee is correct, and an additional contribution from the new surface hybridization adds to the bulk density of states, which induces a sudden enhancement of the amplitude of peak iii).

In order to make this more explicit in our manuscript we have added a sentence in the first paragraph of the *Temperature dependence* part of the Results section: “Following the interpretation of [45] the peak position in energy can be related to the gap edge, i.e. the $\Gamma_8^{(1)}$ band in case of SmB₆, as discussed above.” (here, [45] is the reference to Maltseva, Dzero and Coleman to which the referee refers).

With kind regards on behalf of the authors